# Use of a Smartphone-Based Device for Fundus Examination in Birds: A Pilot Study

**DOI:** 10.3390/ani12182429

**Published:** 2022-09-15

**Authors:** Aure-Eline Grillot, Thomas Coutant, Eva Louste, Cécile Le Barzic, Pascal Arné, Guillaume Payen, Minh Huynh

**Affiliations:** 1Department of Ophthalmology, Centre Hospitalier Vétérinaire Frégis, 43 Avenue Aristide Briand, 94110 Arcueil, France; 2Department of Ophthalmology, École Nationale Vétérinaire d’Alfort, 7 Avenue du Général de Gaulle, 94700 Maisons-Alfort, France; 3Department of Exotic Medicine, Centre Hospitalier Vétérinaire Frégis, 43 Avenue Aristide Briand, 94110 Arcueil, France; 4Departement of Exotics Medicine, École Nationale Vétérinaire d’Alfort, 7 Avenue du Général de Gaulle, 94700 Maisons-Alfort, France; 5Centre Hospitalier Universitaire Vétérinaire de la Faune Sauvage (Chuv-FS), Ecole Nationale Vétérinaire d’Alfort, 7 Avenue du Général de Gaulle, 94700 Maisons-Alfort, France; 6Equipe Dynamyc, UPEC, ANSES, ENVA, 94010 Créteil, France

**Keywords:** fundus imaging, birds, smartphone-based device, Peek Retina, tawny owls, rock pigeons, chickens

## Abstract

**Simple Summary:**

Eye examination is crucial for therapeutic plans and rehabilitation of birds in wildlife rehabilitation centers. However, fundus examination using classical direct or indirect ophthalmoscopy techniques can be challenging in those species. The aim of the study was to assess the use of a smartphone-based retinal imaging system in birds. Fundus examination was feasible in most bird species examined in this study. The difficulties of carrying out the examination seem to be related to the form of the globe, the color of the iris, and the quality of pupil dilation. Further investigations are necessary to confirm these findings.

**Abstract:**

Ophthalmic examination is essential in the avian triage process in order to apply prompt therapeutic plans and evaluate rehabilitation potential. Fundoscopy is traditionally performed by direct or indirect ophthalmoscopy. Recent technological developments have enabled the design of a small-sized and affordable retinal imaging system to examine the fundus. We investigate the use of a smartphone-based device to realize fundus examination through a prospective cross-sectional observational study. Seventy-seven eyes of 39 birds of 15 different species were evaluated using the smartphone-based device in a rescue wildlife center. Pupil dilation was achieved prior to examination via rocuronium topical application. Assessment of fundus by the smartphone was classified as satisfactory, moderately satisfactory, and unsatisfactory. Fundus examination was also performed with a 20D, 30D, or 78D lens for comparison. Pupillary dilation was satisfactory, moderately satisfactory, or absent in 17, 32, and 28 eyes, respectively. Fundus examination with the smartphone-based device was satisfactory, moderately satisfactory, or unsatisfactory in 44, 15, and 18 eyes, respectively. The feasibility of the fundus examination was affected by the form of the globe; by the quality of pupil dilation; by the color of the iris (images could not be obtained from species with an orange, bright iris); and by the species, with owls (Strigiformes) being the easiest to observe. Based on these findings, fundus examination was feasible in most bird species examined in this study.

## 1. Introduction

Ocular lesions are frequently observed in wild birds. Up to 14.5% of birds and 48.1% of raptors admitted to wildlife centers may have ocular lesions [1,2]. In particular, traumatic causes are well documented in raptors [3,4]. Therefore, a complete ophthalmologic examination is recommended when these animals arrive at the rehabilitation center [5]. Specifying the nature and severity of the lesions is important in order to adapt the care, plan rehabilitation protocols, and determine if the animal can be released [6,7]. Ophthalmological examination of birds is more difficult than that of mammals. The presence of voluntarily controlled striated musculature in the iris often makes visual assessment and fundus examination difficult. Moreover, a reliable examination of the fundus traditionally requires specialized equipment operated by experienced veterinarians or veterinary ophthalmologists, who are rarely present on site. Classically, fundus examination in birds is performed using direct or indirect ophthalmoscopy techniques after inducing mydriasis with a neuromuscular blocker, such as rocuronium bromide [8,9,10,11,12,13,14]. Recent technological developments have enabled the design of small, affordable biomedical imaging devices [15]. Several systems are available for the visualization of the fundus [16]. Peek Retina^®^ is one such device, which was selected for this study. It is compatible with any smartphone and has been previously evaluated in humans [16,17,18]. Our anecdotal experience on a few pet birds in a clinical setting (privately owned parrots and pigeons) indicated that fundus examination was very variable among individuals. Especially, we felt that characteristics such as the size of the pupil, the form or the eye itself, or the color of the iris may have an influence on image acquisition. Therefore, we hypothesized that fundus examination using the Peek Retina^®^ could vary in relation to the species, the quality of pupil dilation, the form of the eye, and the color of the iris. This prospective cross-sectional observational study aimed to evaluate the feasibility and the quality of fundus examination using Peek Retina^®^ in different avian species and determine if the examination was more difficult in some species than others.

## 2. Material and Methods

### 2.1. Ophthalmic Examination and Pupil Dilation

Birds were recruited from a wildlife center (CHUV-Faune Sauvage, ENVA, Maisons-Alfort, France). All of the eyes were examined using a slit lamp (Kowa SL 15; Kowa, Dusseldorf, Germany). The difficulty in examining small-sized birds was anticipated. First of all, a very small-sized eye is difficult to examine with direct or indirect ophthalmoscopy. Recommended aspherical lenses for small birds require a 90 D lens, which is not widely used and was indeed not available for our experiment [19]. The second reason was the handling of wild small birds. The handling should be as fast as possible as it can be detrimental as a result of stress-related physiologic response [20]. Therefore, the inclusion criteria were a reasonable size (>200 g) and good tolerance for stress during handling, showing limited or no escape movement. Only eyes with transparent ocular medium were included in this study. Pupils were dilated using a single topical application of 0.20 mg/20 µL rocuronium bromide. Birds were examined in a dark room 45–60 min after rocuronium bromide application. The quality of dilation was subjectively noted as satisfactory, moderately satisfactory, and absent. The effectiveness of dilation was assessed by observing the pupil’s residual capacity to contract.

### 2.2. Fundus Examination

For each bird, fundus evaluation was performed by an ECVO resident (AEG) using a binocular indirect ophthalmoscopic examination with 20 D, 30 D, or 78 D lenses (Volk, Mentor, OH, USA). A smartphone-based retinal imaging device (Peek Retina^®^, Berkhamsted, UK) examination was performed by a second operator (ECZM Avian Resident, TC).

### 2.3. Photographic Equipment

The device technology used is briefly summarized. The Peek Retina^®^ device comprises prisms, an LED, and a power source. The amount of light used to illuminate the retina can be changed by choosing one of three levels of light. The optic device was manually aligned with the smartphone camera to allow fundus imaging and attached using a universal clip. Fundus examination by photography and video recording was performed using a smartphone (Redmi Note 5, Xiaomi, China). The camera application integrated with the smartphone was used for image capturing and manual setting adjustments such as autofocusing, brightness, and clarity. The device was held approximately 10 cm from the subject’s eye and then moved closer. The distance between the smartphone and cornea varied from 2 cm to 6 cm, depending on the position on which the fundus could be focused (Figure 1). The examination began by locating the pecten. The angle of the device was then adjusted to view as much of the fundus as possible. The light level was initially at a lower intensity and was then increased to obtain the best image quality.

Assessment of the fundus was categorized as satisfactory, moderately satisfactory, or unsatisfactory. The first category (satisfactory) corresponds to the capturing of interpretable images of the fundus of good quality within a few seconds after positioning the device in front of the animal’s eye. The pecten as well as the fundus periphery could be observed. The second category (moderately satisfactory) corresponds to the capturing of images of the fundus, requiring multiple adjustments of the position, distance, and angle of the device and images, which made it possible to identify the pecten, but which could lack contrast and sharpness. The third category (unsatisfactory) corresponds to the absence of interpretable imaging of the fundus after multiple attempts.

Anecdotally, we have used the Peek Retina^®^ device on some pet birds and we noticed that the fundus acquisition was variable among species. Therefore, we hypothesized that fundus examination using the Peek Retina^®^ could vary in relation to the species, the form of the eye, and the color of the iris.

Birds’ eyes were categorized as tubular, globoid, and flat shapes, as described in the literature [21,22,23]. The color of the eye was categorized as dark brown, brown, grey, blue, beige, yellow, and orange (Figure 2. As the trial examination of privately-owned birds was not satisfactory with orange irises, a second classification was carried out. The color was then classified as bright or dull. The colors classified as dull were dark brown, brown, grey, blue, and beige. Yellow and orange irises were classified as bright (Figure 2).

### 2.4. Statistical Evaluation

A cumulative odds ordinal logistic regression with proportional odds was run to determine the effect of eye shape, pupillary dilation quality, and iris color on the quality of the fundus assessment using the Peek Retina^®^. In this analysis, the parameters for each eye (left and right of each bird) were considered as independent. Odds ratios were reported with 95% confidence intervals, and values of *p* < 0.05 were considered statistically significant. All statistical tests were performed using the IBM SPSS Statistics software (IBM Corp. Released 2017. IBM SPSS Statistics for Windows, Version 25.0. IBM Corp., Armonk, NY, USA).

## 3. Results

There were 77 eyes of 39 birds included in the study. The birds included 12 Tawny owls (*Strix aluco*), 6 rock pigeons (*Columbia livia*), 4 hens (*Gallus gallus domesticus*), 3 common buzzards (*Buteo buteo*), 2 carrion crows (*Corvus corone*), 2 hobby falcons (*Falco subbuteo*), 1 honey buzzard (*Pernis apivorus*), 1 barn owl (*Tyto alba*), 1 mallard duck (*Anas plathyrhynchos*), 1 Eurasian sparrowhawk (*Accipiter nisus*), 1 herring gull (*Larus argentatus*), 1 grey heron (*Ardea cinerea*), 1 magpie (*Pica pica*), 1 domestic goose (*Anser anser domesticus*), 1 mute swan (*Cygnus olor*), and 1 northern lapwing (*Vanellus vanellus*). The magpie had only one eye examined as the other eye had sequelae of perforation and posterior synechiae, preventing fundus visualization. The eye shapes and iris colors of each species are summarized in Table 1.

Twenty-six eyes of 13 birds, 32 eyes of 16 birds, and 19 eyes of 10 birds were tubular, flat, and globoid, respectively. Five eyes were classified as dark brown, 42 eyes were classified as brown, 4 as grey, 2 as blue, 6 as beige, 4 as yellow, and 14 as orange (4 hens and 3 pigeons).

No ocular or systemic side effects were observed in any of the animals throughout the study. Only mild protrusion of the third eyelid was observed immediately after rocuronium application in all eyes. Pupil dilation was absent in 29 eyes of 16 birds, moderately satisfactory in 29 eyes of 16 birds, and satisfactory in 19 eyes of 12 birds (Table 2). Dilation was not systematically symmetrical between the two eyes of the same bird.

Examination with lenses was possible in all birds, but difficult to achieve and interpret in one carrion crow and one adult pigeon. The fundus was dark in both birds. Lesions were identified in three birds. One tawny owl had hypopigmented and hyperpigmented lesions in the right fundus. One hobby falcon had an abnormal right pecten with a marked decrease of its size and disappearance of pleats. In these cases, the lesions were suggestive of traumatic sequelae. One tawny owl had heterogeneous color of right and left fundi. The sequelae of chorioretinitis were suspected.

Fundus examination with the Peek Retina^®^ device was satisfactory in 48 eyes of 24 birds, moderately satisfactory in 11 eyes of 6 birds, and unsatisfactory in 18 eyes of 9 birds (Table 3). The Peek Retina^®^ did not allow to have an image of the entire fundus on a single sharp image. A video was needed to visualize it (Appendix A). The best images were obtained with the light set to the highest level in all cases. The quality of fundus examination was similar in both eyes of the same bird. It was impossible to obtain an interpretable image of the eye fundi for all four hens, three of the six pigeons, both carrion crows, and the mallard duck. The pigeons for which the fundus was not observable were all adult pigeons with bright orange irises. Evaluation of the fundus was moderately satisfactory in both hobby falcons, the Eurasian sparrow hawk, lapwing, magpie, and herring gull. Fundus examination was simple to perform and of good quality in the other cases. These cases included three of the six pigeons, the tawny owls, common buzzards, hobby falcons, honey buzzard, barn owl, grey heron, domestic goose, and mute swan (Figure 3). The pigeons in this second group were juveniles with brown dull irises.

Among eyes for which pupil dilation was absent and moderately satisfactory, fundus examination with the smartphone-attached device was categorized as satisfactory in 15 of the 29 eyes and 18 of the 29 eyes, respectively.

All fundus lesions visible with converging lenses were also observed using the Peek Retina^®^ device (Figure 4 and Figure 5).

In the cumulative odds ordinal logistic regression with proportional odds to determine the effect of eye shape, pupillary dilation quality, and iris color on the quality of the fundus assessment using the Peek Retina^®^ device, the assumption of proportional odds was met, as assessed by a full likelihood ratio test comparing the fit of the proportional odds model to a model with varying location parameters (χ^2^(4) = 3.8, *p* = 0.43). The final model statistically significantly predicted the dependent variable over and above the intercept-only model (χ^2^(4) = 53.4, *p* < 0.001). Eye shape, iris color, and pupillary dilation statistically significantly predicted the quality of the Peek Retina^®^ fundus assessment (Wald-χ^2^(2) = 8.3, *p* = 0.016; Wald-χ^2^(1) = 14.2, *p* < 0.001; and Wald-χ^2^(1) = 5.1, *p* = 0.024, respectively). More specifically, the odds of the presence of bright iris allowing the assessment of bird’s fundus with the Peek Retina^®^ device was 0.06 (95% CI, 0.01 to 0.26) times that for the presence of a dull iris. Therefore, the presence of a bright iris significantly diminished the quality of the fundus assessment using the Peek Retina^®^ device. The “bright’ category included the yellow iris color of heron and sparrowhawk and orange iris color of adult pigeons and chickens. It is of note that the fundus assessment with the Peek Retina^®^ device was possible with variable quality in all eyes with a yellow iris, but in none of the eyes with an orange iris. The odds of globular eyes allowing the assessment of bird’s fundus with Peek Retina^®^ was 0.06 (95% CI, 0.01 to 0.56) times that of tubular eyes—a statistically significant effect (χ^2^(1) = 6.1, *p* = 0.014). The odds of flat eyes allowing the assessment of bird’s fundus with Peek Retina^®^ was 0.04 (95% CI, 0.01 to 0.35) times that of tubular eyes—a statistically significant effect (χ^2^(1) = 8.3, *p* = 0.004). Therefore, the presence of a globular eye and a tubular eye significantly diminished and increased the quality of the fundus assessment using the Peek Retina^®^ device, respectively. The odds of the quality of pupillary dilation allowing the assessment of bird’s fundus with the Peek Retina^®^ device was 2.4 (95% CI, 1.1 to 5.1). Therefore, the quality of pupillary dilation significantly increased the quality of the fundus assessment using the Peek Retina^®^ device.

## 4. Discussion

Fundus evaluation usually relies on optimal pupillary dilation to observe the posterior segment. Rocuronium bromide has been shown to induce mydriasis in several bird species, including pigeons (*Columbia livia*), common buzzards (*Buteo buteo*), scops owls (*Otus scops*), little owls (*Athene noctuae*), tawny owls (*Strix aluco*), and European kestrels (*Falco tinninculus*) [8,9,10,13,14]. In this study, the effectiveness of dilation was assessed by observing the pupil’s residual capacity to contract. This evaluation of pupil dilation was subjective because the scope of the study was not the evaluation of rocuronium efficacy. These findings may be related to species’ differences and intra-specific variability. In pigeons, iris color is suspected to influence the response to rocuronium bromide application [13]. As the same vial was used in this experiment, there is no possibility of an altered formulation, although some birds responded positively to the application. The examinations were carried out between 45 and 60 min after the application of rocuronium because dilation of the pupil was reported at the end of this period in the species studied [8,9,13,14]. However, pupillary dilation was not necessarily maximal in all individuals. Indeed, maximal pupil dilation in the common buzzard (*Buteo buteo*) has been reported 110 min after instillation [8]. A dose effect may partly explain variations observed between diurnal and nocturnal species, as described by Barsotti [24]. Despite this discrepancy, fundus examination could still be performed using conventional ophthalmoscopy in all birds. For example, the honey buzzard showed inconsistent dilation after rocuronium bromide application, but the fundus was easily examined using both techniques, probably because of the size of the eye itself. In our study, pupillary dilation quality predicted the quality of the Peek Retina^®^ fundus assessment. The optimal use of the Peek Retina^®^ device was thus improved with more adequate pupillary dilation.

Examination using the Peek Retina^®^ device was satisfactory in 48 of the 77 eyes. Some species can be readily evaluated, such as tawny owl, common buzzard, juvenile pigeon, hobby falcon, honey buzzard, barn owl, grey heron, domestic goose, and mute swan; whereas this cannot be evaluated in hens, adult pigeons, carrion crows, and the mallard duck. As expected, eye shape was significantly correlated with a better fundus image quality using the Peek Retina^®^ device. The examination was more qualitative in the case of tubular eyes. Notably, these eyes were the largest in this study. The size of the ocular globe in these cases may have compensated for incomplete mydriasis compared with birds with smaller eyes. Regarding our hypothesis, we thought that the form of the eye would be easier to categorize than measuring the actual size. Measuring the eye accurately (or the corneal diameter) was beyond the scope of the study and would have required either ultrasonographic method or a scale system, which we did not implement at the time of the procedure. This would be a potential parameter to integrate into further studies.

In situations where the device could not capture an image, the autofocus of the smartphone was unable to focus on the fundus. A diffuse gray blur was most often observed. This could be related to the focal distance of the retina in these species or a noise component in the image signal [25]. We found that the iris color significantly influenced the quality of the fundus image using the Peek Retina^®^ device. Indeed, a lack of contrast may enhance the effect of noise on the image [26]. It could be postulated that a black fundus with a black iris in carrion crows, or a red iris with red eyelids in chickens, may affect the smartphone autofocus function. Interestingly, fundus examination was easier to perform in young pigeons than in adult pigeons. Young pigeons had dark irises; either a non definitive color that become brighter as they age or a specific eye color encountered in pigeons («bull eye» color) [27]. Having a dull pigmented iris may have facilitated the accommodation of the smartphone device. Most species in Europe have a dull iris [28], as seen in our cohort. It is of note that fundus assessment with the Peek Retina^®^ device was acceptable or satisfactory in all eyes with a yellow iris, but in none of the eyes with an orange iris. The degree of brightness as well as the red coloration part of orange iris may have impacted the focus of the device. Brightness is reported to influence the autofocus algorithm and may lock the image on the brighter element, especially the red pigment [29]. Because image acquisition also relies on a smartphone camera, it is unknown whether a more recent smartphone equipped with a better photographic algorithm would be able to obtain more qualitative images. Finally, the examiner’s progress curve could have an impact on the results.

The use of smartphone-based fundoscopy has been evaluated in other species such as cats, dogs, and rabbits [30]. The device allowed the evaluation of the optic nerve, tapetum lucidum, non-tapetal region, retinal vessels, and choroidal vessels. Some anatomical differences are expected between birds. The retina is anangiotic and clearly visible, and the optic nerve is largely masked by the pecten and is only seen as a narrow white margin at the base of the pecten [21,22,23]. In our group, the pecten and retinae could be clearly seen whenever possible.

Ocular lesions are described in different species admitted to wildlife rehabilitation centers and consist of retinal detachments or tears (71% according to Moore et al., 2017) [31]. They also include retinal discoloration, posterior uveitis, pecten lesions, and chorioretinitis [4,5,32]. The smartphone device detected retinal lesions in three birds that were affected in our group. Smartphone-based retinal cameras have been successfully used in human medicine to screen for diabetic retinopathy, and several devices have been compared for the fundus examination [16]. The performance of the Peek Retina^®^ device was similar to that of the D-eye^®^, but had a lower field of view compared with the iNview^®^. Only 45% of the human fundus was visible in each image using the Peek Retina^®^ device. These results were similar to the present study in that a video and several images were necessary in order to reconstruct the full fundus image, which allowed better visualization and understanding of the fundus.

## 5. Conclusions

This study is the first to evaluate the use of a smartphone-based camera system for fundus evaluation of birds. Although the device shows promising prospects, such as immediate archiving, sharing options, and potential use by volunteers, several limitations were encountered. Iris color and brightness may have altered capacity of the device to obtain sharp images. The shape of the eye, pupil diameter, and probably eye size may have impacted the examination of the fundus. The device seems to be practical and easy to use in Strigiformes and potentially in some other species. Further studies should be conducted on other avian orders in order to confirm the factors that may influence the use of the device.

## Figures and Tables

**Figure 1 animals-12-02429-f001:**
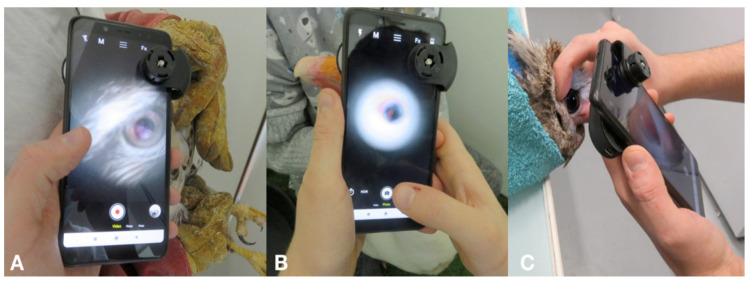
Fundus examination using the Peek Retina^®^ device of (**A**) a common buzzard (*Buteo buteo*); (**B**) a herring gull (*Larus argentatus*); (**C**) a tawny owl (*Strix aluco*) (Appendix A).

**Figure 2 animals-12-02429-f002:**
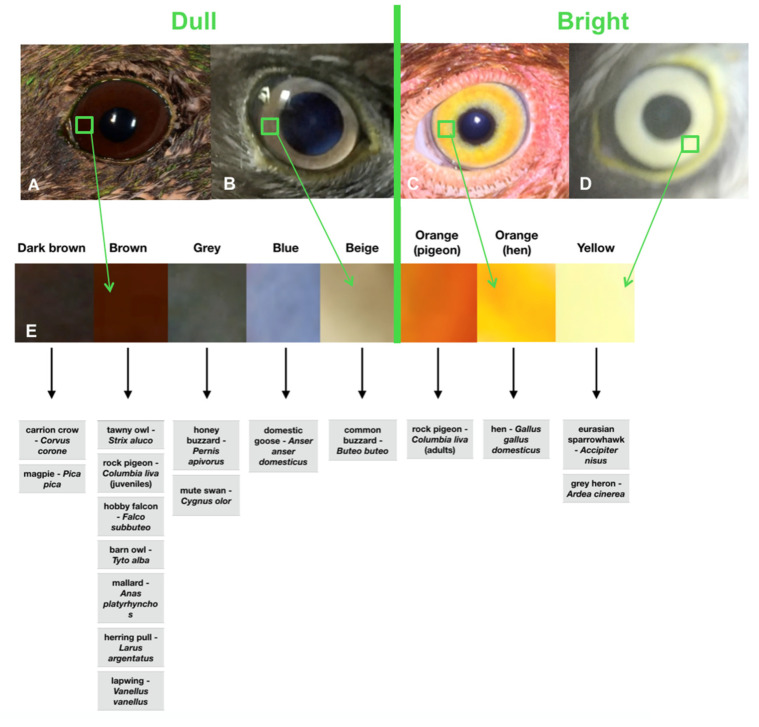
(**A**) Eye of a mallard duck (*Anas plathyrhynchos*); brown coloration of the iris was classified as dull. (**B**) Eye of a common buzzard (*Buteo buteo*); beige coloration of the iris was classified as dull. (**C**) Eye of a hen (*Gallus gallus domesticus*); orange coloration of the iris was classified as bright. (**D**) Eye of an Eurasian sparrowhawk (*Accipiter nisus*); yellow coloration of the iris was classified as bright. (**E**) Iris color scale of irises encountered.

**Figure 3 animals-12-02429-f003:**
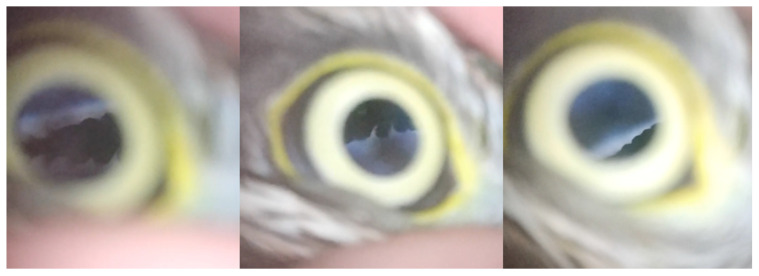
Fundus of an Eurasian sparrowhawk (*Accipiter nisus*) showing pecten normal aspect. This fundus examination was classified as satisfactory.

**Figure 4 animals-12-02429-f004:**
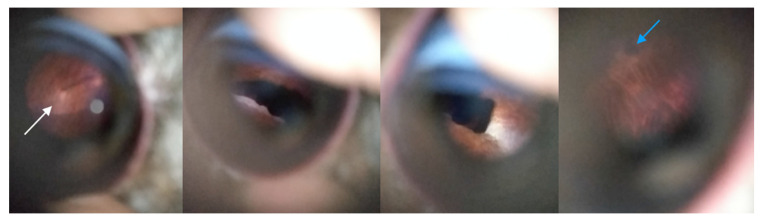
Fundus of a tawny owl (*Strix aluco*) showing a hypopigmented zone (white arrow) and a hyperpigmented zone (blue arrow). Sequellae of trauma were suspected. This fundus examination was classified as satisfactory.

**Figure 5 animals-12-02429-f005:**
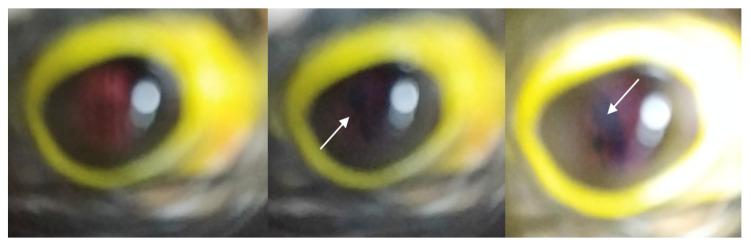
Fundus of a hobby falcon (*Falco subbuteo*) showing an anomaly of the pecten (white arrow). A degenerative process secondary to eye trauma was suspected. This fundus examination was classified as acceptable as fundus components were identified, but the examination was not totally satisfactory because of blur and light reflection.

**Table 1 animals-12-02429-t001:** Shape of the globe and iris coloration of avian species included in the study.

Species (Number of Animals)	Order	Family	Shape of the Globe	Iris Coloration
tawny owl—*Strix aluco* (12)	Strigiformes	Strigidae	Tubular	Brown
rock pigeon—*Columbia liva* (6)	Columbiformes	Columbidae	Flat	Bright orange (Adults); Brown (Juveniles)
hen—*Gallus gallus domesticus* (4)	Galliformes	Phasianidae	Flat	Bright orange
common buzzard—*Buteo buteo* (3)	Acciptriformes	Accipitridae	Globoid	Beige
carrion crow—*Corvus corone* (2)	Passeriformes	Corvidae	Globoid	Dark brown
hobby falcon—*Falco subbuteo* (2)	Falconiformes	Falconidae	Globoid	Brown
barn owl—*Tyto alba* (1)	Strigiformes	Tytonidae	Tubular	Brown
honey buzzard—*Pernis apivorus* (1)	Acciptriformes	Accipitridae	Globoid	Grey
mallard—*Anas platyrhynchos* (1)	Anseriformes	Anatidae	Flat	Brown
eurasian sparrowhawk—*Accipiter nisus* (1)	Acciptriformes	Accipitridae	Globoid	Yellow
herring pull—*Larus argentatus* (1)	Charadriformes	Laridae	Flat	Brown
grey heron—*Ardea cinerea* (1)	Pelecaniformes	Ardeidae	Flat	Yellow
domestic goose—*Anser anser domesticus* (1)	Anseriformes	Anatidae	Flat	Blue
lapwing—*Vanellus vanellus* (1)	Charadriformes	Charadridae	Flat	Brown
magpie—*Pica pica* (1)	Passeriformes	Corvidae	Globoid	Dark brown
mute swan—*Cygnus olor* (1)	Anseriformes	Anatidae	Flat	Grey

**Table 2 animals-12-02429-t002:** Quality of pupil dilation 45 to 60 min after rocuronium instillation in both eyes in the birds included in the study.

Species (Number of Animals)	Satisfactory (n/Total Eye Number per Species)	Acceptable (n/Total Eye Number per Species	Absent (n/Total Eye Number per Species
tawny owl—*Strix aluco* (12)	7/24 eyes	10/24 eyes	7/24 eyes
rock pigeon—*Columbia livia* (6)	1/12 eyes	5/12 eyes	6/12 eyes
hen—*Gallus gallus domesticus* (4)	-	4/8 eyes	4/8 eyes
common buzzard—*Buteo buteo* (3)	2/6 eyes	2/6 eyes	2/6 eyes
carrion crow—*Corvus corone* (2)	-	-	4/4 eyes
hobby falcon—*Falco subbuteo* (2)	2/4 eyes	2/4 eyes	-
barn owl—*Tyto alba* (1)	-	2/2 eyes	-
honey buzzard—*Pernis apivorus* (1)	-	-	2/2 eyes
mallard—*Anas platyrhynchos* (1)	1/2 eyes	-	1/2 eyes
eurasian sparrowhawk—*Accipiter nisus* (1)	1/2 eyes	1/2 eyes	-
herring pull—*Larus argentatus* (1)	-	2/2 eyes	-
grey heron—*Ardea cinerea* (1)	2/2 eyes	-	-
domestic goose—*Anser anser domesticus* (1)	2/2 eyes	-	-
lapwing—*Vanellus vanellus* (1)	-	-	2/2 eyes
magpie—*Pica pica* (1)	-	-	1/1 eye
mute swan—*Cygnus olor* (1)	1/2 eyes	1/2 eyes	-
39	19	29	29

**Table 3 animals-12-02429-t003:** Quality of fundus examination of birds examined with the Peek retina^®^ device.

Species (Number of Animals)	Satisfactory (n/Total Eye Number per Species)	Acceptable (n/Total Eye Number per Species)	Unsatisfactory (n/Total Eye Number per Species)
tawny owl—*Strix aluco* (12)	24/24 eyes	-	-
rock pigeon—*Columbia livia* (6)	6/12 eyes	-	6/12 eyes
hen—*Gallus gallus domesticus* (4)	-	-	8/8 eyes
common buzzard—*Buteo buteo* (3)	6/6 eyes	-	-
carrion crow—*Corvus corone* (2)	-	-	4/4 eyes
hobby falcon—*Falco subbuteo* (2)	-	4/4 eyes	-
barn owl—*Tyto alba* (1)	2/2 eyes	-	-
honey buzzard —*Pernis apivorus* (1)	2/2 eyes	-	-
mallard—*Anas platyrhynchos* (1)	2/2 eyes	-	-
eurasian sparrowhawk—*Accipiter nisus* (1)	2/2 eyes	-	-
herring pull—*Larus argentatus* (1)	-	2/2 eyes	-
grey heron—*Ardea cinerea* (1)	-	2/2 eyes	-
domestic goose—*Anser anser domesticus* (1)	2/2 eyes	-	-
lapwing—*Vanellus vanellus* (1)	-	2/2 eyes	-
magpie—*Pica pica* (1)	-	1/1 eye	-
mute swan—*Cygnus olor* (1)	2/2 eyes	-	-
39	48	11	18

## Data Availability

The data presented in this study are available in Table 1, Table 2 and Table 3.

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
