# Peer review of "Use of a Smartphone-Based Device for Fundus Examination in Birds: A Pilot Study"

_animals, 2022, doi:10.3390/ani12182429_

Round 1

Reviewer 1 Report

It is necessary to correct  the word ophtalmic --> ophthalmic

Reviewer 2 Report

Birds presented at veterinary practices and wildlife rehabilitation centres are often suffering from eye trauma and so when dealing with avian patients an ophthalmic assessment should always be undertaken if possible. However, as the authors point out, this is not always straightforward as it can require specialist equipment and/or knowledge and skills. Modern, easy-to-use and affordable technology that can be used in conjunction with smartphones offers the potential for any vet/volunteer to perform at least a cursory (but nevertheless important) eye examination in an avian patient. In this study the authors assess the feasibility of just such an approach. I think this is an important and interesting paper that highlights both advantages (and limitations) of using modern techniques to perform fundus examinations in birds. However I have some concerns that I feel need to be addressed before I am prepared to recommend this manuscript for publication.

Specific points

Abstract

Check spelling of ophthalmic and ophthalmology

I think the number of species examined should be include as well as the number of individual birds and eyes

Change “The pupil dilation was prior achieved” to Pupil dilation was achieved prior to examination”.

Change “(Strigiformes being the easiest one to observe)” to “(with owls (Strigiformes) being the easiest to observe)”.

Change “the iris (image could not be obtained with orange bright iris) ” to “the iris (images could not be obtained from species with an orange, bright image)”.

Change “Based on those” to “Based on these”.

Remove “especially in Strigiformes where it provided adequate quality images.” (This repeats what has already been written above.)

Introduction

Change “48,1%” to “48.1”.

Introduction and throughout. Why is the tradename “Peek Retina” sometimes followed by a registered trademark symbol (®) and sometimes not? There should be consistency throughout the text as to how this is presented.

Change “Peek Retina® is one of these devices” to “Peek Retina® is one such device,”.

Change “fundus examination using the Peek Retina” to “fundus examination using Peek Retina”.

Hypotheses. The authors state that their hypotheses were that the use of Peek Retina in some birds would be more difficult due to variation in eye shape and also the colour of the iris. Why? The authors offer no background information in the introduction that would form the basis of either of these hypotheses. What existing information is there that iris colour would influence the effectiveness of Peek Retina, for example? Furthermore, I would have thought that eye size would be a major limitation on the usefulness of this device, yet the authors do not mention eye size (or bird size, as eye size is correlated with body size in birds) other than stating in the methods that birds > 200g were used in this study. Looking at the list of species used, I would estimate that all have an equatorial eye diameter of at least 10 mm, with some species, such as owls and large raptors having much larger eyes. That the authors chose to only use birds >200g indicates that they are aware of this potential limitation with this method, but they do not expand upon this elsewhere in the manuscript. I do not think that this takes anything away from this study, but it seems to me that if Peek Retina does not work well for smaller species (like small garden passerines, for example), then this will be an important result.

Methods

Change “Examination began by locating the pecten, the angle of the device was then adjusted” to “Examination began by locating the pecten. The angle of the device was then adjusted”.

Iris colour. I do not understand the criteria for describing irises as being either bright or dull. I can imagine that an orange iris can subjectively called ‘bright’ and that a brown or grey iris could be termed ‘dull’, but what about a yellow iris? I have seen birds with yellow irises that I would subjectively consider ‘bright’, especially when compared to a brown or grey iris. And subjectively a pale blue iris can also appear bright to some viewers. The authors have provided definitions for each of the categories of fundus assessment, and I think they also need to provide more robust definitions of how they determined whether iris colour was either bright or dull as well. At present this seems very subjective and potentially non-repeatable. Perhaps including photographs of all of the different iris colours would help, rather than just presenting the two examples shown in figure 2.

Results

When presenting the results, I would have expected the number of species examined to feature as one of the first pieces of information presented. I also feel that a lot of the first paragraph of the results section is unnecessary as it repeats the information presented in table 12.

Table 1. I think table 1 could be improved by providing additional taxonomic information for each species, e.g. order and family. The authors refer to Strigiformes throughout the text so having this information provided for all specie sin table 1 would be useful.

Statistical analysis. I have not commented on the statistical analysis regarding iris colour because I am assuming that the authors will be able to justify their bright and dull criteria for iris colour.

Figure 5 legend. Change “blurr” to “blur”.

Discussion and Conclusions

When discussing why the eyes of some species could be more easily examined than others, the authors state that they expected eye shape to play a role, but again do not state why? What are the reasons behind their hypothesis that eye shape is correlated with a better fundus image? Also, what about the influence of eye size? The authors make no mention of whether, in general, birds with bigger eyes were easier to assess, other than the owls with their very large tubular eyes.

The authors state that the examination was more qualitative in the case of tubular eyes. What does this mean?

In the paragraph about the influence of iris colour, the authors provide some information about how smartphone autofocus mechanisms may be influenced by brightness. It seems to me that this is the kind of background information that should have been presented in the introduction as well, so the reader can understand how the authors came to generate their hypothesis about iris colour being an important variable in determining how well the use of Peek Retina would be in assessing the avian fundus.

Change “Europe have a dull iris 24” to “Europe have a dull iris [24]”.

By only selecting birds > 200g, the authors seem to have already determined that the use of a smartphone-based camera system may only work in larger birds and/or birds with eyes above a certain size, and yet they do not mention this in the conclusions or in the discussion. I think the use of smartphone-based camera systems have fantastic potential application in veterinary ophthalmology, but it is important that we fully understand their limitations.

Round 2

Reviewer 1 Report

Good afternoon

thanks for your corrections..can I ask to consider the corrections only and delete the crossed red old sentences? in order to be sure how looks the final version? 

Is there anything else to do  for the quality of the images?

thanks a lot

Author Response

Thank you very much for your comments

- can I ask to consider the corrections only and delete the crossed red old sentences? in order to be sure how looks the final version? 

There is apparently a file conversion problem. Indeed, corrections and deletions are reversed. The most recent corrections appear crossed out. We have returned a new word document and a pdf version if it doesn’t work. 

- Is there anything else to do  for the quality of the images?

About the photographs, they are captures of videos. The aspect of the fundus is more appreciable in video format than in still images. Structures are better perceived with movement, especially the relief of the pecten. The only way to improve the situation would be to add an additional document video. We proposed it to the editor.
